# The Evaluation of Winter Wheat Adaptation to Climate Change in the Central Non-Black Region of Russia: Study of the Gene Pool Resistance of Wheat from the N.I. Vavilov Institute of Plant Industry (VIR) World Collection to Abiotic Stress Factors

**DOI:** 10.3390/plants10112337

**Published:** 2021-10-29

**Authors:** Sulukhan K. Temirbekova, Ivan M. Kulikov, Yuliya V. Afanasyeva, Olga O. Beloshapkina, Elena A. Kalashnikova, Rima N. Kirakosyan, Peter A. Dokukin, Dmitry E. Kucher, Mourad Latati, Nazih Y. Rebouh

**Affiliations:** 1All-Russian Research Institute of Phytopathology, Bolshye Vyazyomy, Odintsovo District, 143050 Moscow, Russia; sul20@yandex.ru; 2Federal Horticultural Center for Breeding, Agrotechnology and Nursery, 115598 Moscow, Russia; vstisp@vstisp.org (I.M.K.); yuliya_afanaseva_90@bk.ru (Y.V.A.); 3Moscow Timiryazev Agricultural Academy, Agrarian University, 127550 Moscow, Russia; beloshapkina58@mail.ru (O.O.B.); kalash0407@mail.ru (E.A.K.); r.kirakosyan@rgau-msha.ru (R.N.K.); 4Peoples’ Friendship University of Russia (RUDN University), 6 Miklukho-Maklaya Street, 117198 Moscow, Russia; p.dokukin@gmail.com (P.A.D.); kucher-de@rudn.ru (D.E.K.); 5Ecole Nationale Supérieure Agronomique (ES1603), Laboratoire d’Amélioration Intégrative des Productions Végétales (C2711100), Département de Productions Végétales, Avenue Hassane Badi, El Harrach, Algiers 16200, Algeria; m.latati@yahoo.com

**Keywords:** abiotic stress, climate change, wheat breeding, wheat resistance, VIR gene pool

## Abstract

The paper presents the results of a 50-year research of the genepool of the winter wheat from the world’s largest wheat collection of N.I. Vavilov Institute of Plant Industry (VIR) to investigate its resistance to the abiotic stress factors of the Moscow region and see how closely it matches the attributes of a wheat ideotype as postulated by N.I. Vavilov in 1935. The critical years in studying the wheat’s winter resistance were 10 years out of 50: excessive water saturation during the year 2013; soil drought in 1988; and atmospheric drought in 1972 and 2010. During the investigation, the following gene pool features were analyzed: frost characterized by the cultivar Sojuz 50 (Russia), rapid temperature change, thawing, ice, and rotting resistance characterized by the cultivars Zarya 2 (Russia), Sv 75268, (Sweden), Caristerm and Tukan (Germany), PP 114-74 and Liwilla (Poland), Maris Ploughman and Granta (Great Britain), Titan (USA), Zdar (Czech), and Zenta (Switzerland); regeneration capacity in spring after poor wintering expressed by the cultivars Pamyati Fedina (Russia), TAW 3668.71 (Germany) and Rmo (Poland); resistance to excessive soil and air saturation exhibited by the cultivars Moskovskaya 39 (Russia), Tukan, Compal, Obelisk, Orestis, and Bussard (Germany); solid standing culm that is resistant to lodging characterized by the cultivars Tukan, Kronjuwel, Compal (Germany), Zenta (Switzerland), Moskovskaya 56 (Russia), and Hvede Sarah (Denmark); resistance to enzyme-mycotic depletion of seeds characterized by the cultivars Tukan, Compal, Obelisk, Orestis, Bussard (Germany), Sv 75268, Helge, VG 73394, Salut, Sv 75355 (Sweden), Zenta (Switzerland), Moskovskaya 39, and Ferrugineum 737.76 (Russia); and resistance to soil and atmospheric drought demonstrated by the cultivars Liessau, Heine Stamm, Severin, Neuzucht 14/4, Haynes, Rus 991, Halle 1020 (Germany), Gama (Poland), Sv 71536 (Sweden), and Moskovskaya 39 (Russia). Moreover, the cultivar Mironovskaya 808 (Ukraine) showed resistance to almost all abiotic stress factors studied. The performed study contributes towards the provision of potential sources of resistance to abiotic stress factors prevalent in the Moscow region that can be incorporated in advanced breeding programs.

## 1. Introduction

Abiotic stresses such as frost, rapid temperature change, thawing, ice, and rotting are the major factors affecting the growth and yield-related characters of wheat [1,2]. Wheat yield losses caused by abiotic stresses are around 40% of world wheat crop production [3]. This could directly affect the food security of the burgeoning population, which is projected to increase up to 9.1 billion in 2050 [4,5]. It is necessary to develop wheat varieties that could cope with abiotic stress to meet the daily needs and to achieve food self-sufficiency of this population increase.

Previously carried out research demonstrates that one of the most important ways to increase wheat production is developing new cultivars [6,7]. Developing new wheat varieties adaptable to the soil and the specific climatic conditions in different regions is considered as the key option to increase and sustain wheat production [8]. The use of genetic approaches allows the development of wheat varieties that are tolerant to abiotic stress (i.e., frost and drought) in order to reduce the stress impact on wheat growth and yield [9]. In fact, appropriate selection criteria enable breeders to use the genetic variation for enhancing stress tolerance in crops [10].

For over 50 years, the world winter wheat collection represented by three thousand samples at the Center for the Gene Pool and Bioresources of Plants, formerly the Moscow Branch of VIR, has been used by plant breeders from various research institutes to investigate possible sources of valuable agronomic traits for new cultivars selection. 

N.I. Vavilov [11] noted in his work on ideal cultivars that a plant breeder should clearly understand what qualities will be the focal point of their work. It was found that in the context of winter wheat cultivation in the Central Region of the Russian Federation these primary qualities were frost, rapid temperature change, thawing, ice, and rotting resistances; regeneration capacity in spring after a bad wintering; resistance to excessive soil and air saturation; solid standing culm that is resistant to wind and rainfall; resistance to enzyme-mycotic depletion of seeds; resistance to moisture stress; and resistance to plant diseases, which defines the line of research of winter wheat gene pool in the Russian non-Chernozem Belt.

Winter wheat is among the staple cereal crops in Russia cultivated on extensive area of about 10 million hectares [12]. Wheat provides nearly 55% of the carbohydrates and daily protein for 85% of the world’s population [13], however, abiotic stresses are a serious threat to wheat crop production across the globe. Thus, after evaluating the resistance of winter wheat to biotic stress in our previous study [14], the present paper aims at investigating the gene pool of winter wheat from the world collection of VIR and identify the sources of resistance to the region’s abiotic stress factors for future breeding.

## 2. Methodical Approach Applied in Plant Breeding

The former Moscow Branch of VIR is located in the Moscow Region, which in turn falls within the Central part of the south taiga forest soil-climate zone (the Central Region of the non-Chernozem Belt). The climate is moderately continental and humid. The average annual precipitation is 450–800 mm, and the moisture is sufficient in years with normal precipitation. The probability of excessively wet years is 25–40%, and that of arid and semiarid years is about 12–20%. Accumulated temperatures above 10 °C decrease from 2100° in the east and south-east to 1900° in the north-west, and vegetation periods (above 10 °C) decrease accordingly from 140–45 to 120–125 days [15].

The field experiment was carried out under soddy podzolic soils under an acid conditions pH ranged from 4.5 to 5.6. The humus content was varied from 2 to 2.25 % with a thin humus layer and base-unsaturated exchange complex. The nitrogen, phosphorus, and potassium contents in the soil were, respectively, ranged from 18.5 to 19.5, 9.29 to 10.74, and 17.81 to 19.78 mg/100 g. The podzolic layer was well defined. The underlying rock was represented by moraine loams. Soil erosion was weak.

The water availability and heat resource in the Moscow Region makes it possible to cultivate almost all temperate zone crops. About 70% of precipitation is recorded in the warm season, which ensures favorable conditions for plant growth and development. The case study is located in the Stupino District, which is a part of the second agroclimatic zone delimiting the central part of the region included in subzone 11 a of soddy podzolic loam soils. Soils freeze to the depths of 50–75 cm in open areas and 30–50 cm in sheltered areas. Full thawing of soil is usually expected from 21 to 29 April. Soil tilth is achieved on 20 May in loam soils and on 18 May in sandy loam soils. The frost-free period usually lasts 120–135 days, which allows cultivated plants to achieve full ripeness. Permanent snow cover with an average depth of 35 cm from 25 November to 2 December, persisting up to 137–143 days. The hydrothermal index is 1.3–1.4. North- and south-west are the prevailing wind direction trends in the Moscow Region throughout the year [16].

Winter wheat samples were introduced into a well-defined crop rotation in late August with black fallow crops. An SSFK-7M seeder was used, and the sowing density was 500 grains per 2 m^2^. Mineral fertilizers were applied during the land preparation period with follows rates: 68 kg/ha for nitrogen, 60 kg/ha for phosphorus, and of 30 kg/ha for potassium. Furthermore, 50 kg/ha of nitrogen was applied as top dressing in spring. The reference cultivars Mironovskaya 808 (k-43920, Ukraine) and in some years Polukarlik 3 (k-54508, Ukraine) and Moscow cultivars, such as Zarya (k-49916), Nemchinovskaya 52 (k-59269), Moskovskaya 39 (k-64160), etc., were planted with intervals of 10 and 50 samples, respectively. The wheat collection was studied in compliance with the VIR Methodical guidelines [17,18,19,20] and the International COMECON list of descriptions for genus Triticum L. [21]. 

Every year, 500–700 genotypes of winter wheat were studied, the precursor was complete fallow. The data collected and analyzed annually were germination, overwintering, snow mold damage, frost resistance, regenerative ability of plants in the spring after overwintering, resistance to drought and waterlogging, vegetation period, plant height, lodging, and the weight of 1000 grains. Yield and the grain quality were mesured only for the distinguished varieties. Every year, genotypes distinguished by resistance to abiotic stresses were statistically evaluated with a standard variety based on yield, weight of 1000 grains and plant height.

## 3. Frost Resistance 

The Moroz (Frost) project headed by L.I. Surkova (PhD), which combined scientific efforts of research institutes from different regions, was carried out at the Moscow Branch of VIR [22]. The long-term research showed that the winter wheat collection consistently survived strong frosts at temperatures as low as −20 °C. No damage to the collection has been reported after 50 years. 

## 4. Winter Resistance or Resistance to Rapid Temperature Change, Thawing, Ice, and Rotting

N.I. Vavilov emphasized that breeding for winter resistance is vital for sustainable wheat production [23]. The critical years to study the winter resistance of the winter wheat collection in the Moscow Region were 10 years out of 50.

The winter of 1977/1978 was extremely unfavorable for cereal production. In November, the average temperature was −2.7 °C, i.e., 4 °C above the long-term average, while the precipitation in the form of rain and snow exceeded the norm by 64 mm (110.2 mm). As a result, the soil did not freeze, and the snow cover formed over a thawed soil, which later resulted in the rotting and freezing of the wintering crop. The physiological processes associated with respiration continued throughout winter leading to exhaustion and decimation of plants in winter. The rye collection (300 samples) was destroyed completely, and only 30% of the winter wheat collection (a total of 900 samples) survived. Two reference cultivars Mironovskaya 808 and Zarya exhibited higher resistance as compared to the thinned and empty plots of other wheat collection samples. The survival rate was 65–75%, with Zarya’s rate reaching 75%. The latter also showed high crop yield and was used as a donor for head smut resistance. 

The wintering conditions of 1984/1985 were also critical for winter crops, especially from 10–30 November. The minimum air temperature fell sharply from −3.5 to −23.9 °C. The lack of snow cover resulted in the freezing of plants and ice crust formation at the soil surface, and eventually the loss of samples from Krasnodar Krai, as well as Germany, the Netherlands, Belgium, and France (Figure 1).

In the winter of 1985/1986, 38 collection samples were lost completely, and 82 new samples showed a wintering score of 1. The latter included 15 samples from the United States, 15 from Germany, and 5 from the Netherlands, Poland, Denmark, and Sweden; 102 samples scored 3 (low). There was not a single sample with a score of 9. Among the plants designated for the State Custody, 126 samples were lost completely, and 132 samples showed high plant mortality. Plant rottenness was exacerbated by deep snow cover combined with the lack of soil freezing in the middle of winter. Snow cover was formed over the thawed soil, intense thawing (2–4 °C) was reported in early, middle, and late December with precipitation in the form of rain. Moderate frost (−8–12 °C) was reported in January with snow cover depths of 40–50 cm. The temperature went up to 3 °C once again on the third day of January. Persistent frost in early February did not affect soil freezing due to deep snow cover (50–60 cm). Snow compaction was performed to lower the temperature of the upper soil layer. Despite that, a lengthy period of above-zero temperatures under the snow cover led to plant exhaustion at the end of winter that combined with the cold spell in spring following snow-cover melting had a ruinous effect on many samples. 

The surviving wheat collection samples generally exhibited low crop yields. Reference cultivars Mironovskaya 808 and Zarya turned out to be less resistant to multiple thawing than Swedish, German, and Ukrainian cultivars. The reference cultivars showed crop yields of 240–249 g/m^2^ compared to 345–460 g/m^2^ of the resistant cultivars. The latter included Sv75268 (k-56156), Helge (k-56872), WW23977 (k-56875), Hildur (k-54130), SvVG74393 (k-56065), and Sv 01744 (k-56159) from Sweden; TAW37564 (k-55939) and TAW5127.72 (k-55940) from Germany; and Lvovskaya ostistaya (k-55767, Ukraine).

The winter of 1988/1989 was exceptionally warm with the average monthly ambient temperature in January–March exceeding the norm by 9.1 °C. In this context, a rare phenomenon of lack of soil freezing throughout the whole winter combined with deep snow cover (over 60 cm) was observed. Thawed soils combined with deep snow cover produced compromising conditions for wintering plants. Despite the artificial compaction, the adverse effect of rotting could not be negated. As a result, the winter wheat collection showed significant mortality (over 70 samples), 47 samples were lost in the nursery, and significant mortality due to rotting and snow mold was reported in the winter rye collection.

Resistant samples emerged with crop yields of 450–630 g/m^2^ and 1000 grain weights of 40–47 g compared to reference cultivar Mironovskaya 808 with 365 g/m^2^ and 47 g, respectively. The resistant samples included PP114-74 (k-57618) and Liwilla (k-57580) from Poland; Zdar (UH 7050) (k-57255) from the Czech Republic; Maris Ploughman (k-57944) from the United Kingdom; Remus (k-56904), Caristerm (k-57610), and Tukan (k-57579) from Germany; Salut (k-58035) and Sv VG73394 (k-56160) from Sweden; Raduga (k-50948), Nemchinovskaya 846 (k-56861), Nemchinovskaya 110 (k-56858), and Lyutestsens 497.83 (k-57657) from the Moscow Region, Russia; and Brigantina (k-55181) from Ukraine.

The winter of 1989/1990 was warm, with above-zero temperatures as early as February. The thawed soil had a snow cover of 60 to 70 cm. Frequent thawing in winter led to rotting and snow mold infection. A total of 164 collection samples from Western Europe and Krasnodar Krai were lost. The propagation nursery was most affected by rotting. The following samples stood out in terms of snow mold and winter resistance: Pokal (k-56827) from Austria; Hvede Sarah (k-56289) from Denmark; TAW4279180 (k-58363), Fakta (k-57582), Compal (k-57585), Fakon (k-58187), Kronjuwel (k-57615), and TAW39496.75 (k-56903) from Germany; Venture (k-57231), Longbow (k-57611), and Granta (k-57219) from the United Kingdom; Sv75355 (k-56158) from Sweden; Titan (k-58059) from the United States; Eritrospermum 9736 (k-57479) and Grekum 9271 (k-57472) from Ukraine; Zarya 2 (k-54610) and Nemchinovskaya 52 (k-59269) from the Moscow Region, Russia; and Belosnezhnaya (k-57573) from the Rostov Region. They showed crop yields of 400–600 g/m^2^, compared to 495 g/m^2^ of the reference cultivar Mironovskaya 808. 

The winter of 1992/1993 was extremely unfavorable for winter crops. Rotting and ice resulted in a loss of most collection samples of winter crops; Mironovskaya 808 showed a survival rate of only 40%. 

The winter of 1997/1998 was warm, but the snow cover in January and February was shallow, while the precipitation was two times as low as the normal amount. As a result, wintering crops were significantly damaged by rotting, and wintering was deemed unsatisfactory. After the snow cover melted, rye, wheat, and triticale samples were strongly affected by snow mold infection, so the mortality rate reached 50–70%. 

German cultivars Compal (k-57585) and Tukan (k-57579) and reference cultivars Zarya and Mironovskaya 808 turned out to be the most winter-resistant in these severe conditions. Compal and Tukan showed 1000 grain weights of 39.3–41.5 g and crop yields of 290–350 g/m^2^, the respective values for the reference cultivars being 45–56 g and 330–470 g/m^2^.

The winter of 2000/2001 stood out in terms of thawing and rainfall with an aggregate precipitation of 193.9 mm compared to the normal amount of 111 mm. Weak soil freezing caused intense rotting and snow mold infection leading to the high mortality of winter crop collection samples as well as of perennial grass in many farms in the Moscow Region and other territories. 

Rotting and intense snow mold infection resulted in a loss of 60–80% of the collection samples from Western Europe, CIS, and Russia.

In the winter months of 2002/2003, especially in December, frost with little or no snow was reported. January and February were warm with average monthly ambient temperatures of −6.2 °C and −8.9 °C compared to the norms of −10.8 °C and −9.6 °C, respectively. All these factors combined had a negative effect on wintering crops and perennial grass. Crop loss was caused by ice and snow mold infection. Enzyme-mycotic depletion of seeds was reported in 200 collection samples.

Among the 1200 collection samples analyzed, the following cultivars stood out in terms of crop yields as well as winter and snow mold resistance in 2002–2003: Pamyati Fedina (k-62440) (1993) with crop yield of 420 g/m^2^, Nemchinovskaya 24 (k-65757) (2006) with 450 g/m^2^, Moskovskaya 56 (k-65760) (2008) with 400 g/m^2^, Lyutestsens 319 (k-59267) with 422 g/m^2^, Ivanovskaya 16 (k-58526) from Russia with 367 g/m^2^; Varmlands (k-34230) from Sweden with 302 g/m^2^; Obelisk (k-62032) with 300 g/m^2^, Orestis (k-64034) with 310 g/m^2^, Bussard (k-64027) with 400 g/m^2^, and Gelderseries with 360–490 g/m^2^ from Germany; Zenta (k-56825) from Switzerland with 280 g/m^2^; and reference cultivar Mironovskaya 808 (1963) with 280 g/m^2^.

In 2004/2005, poor wintering of the gene pool of winter crops due to rotting and ice was reported.

Field evaluation of the snow mold infection caused by *Fusarium nivale* Ces. was performed in spring. Severe wintering conditions and further snow mold infection of collection samples led to a loss of 55 out of 67 received samples [24]. 

Among the 528 winter wheat collection samples, the following cultivars stood out: Kazanskaya 560 (k-63565) from Tatarstan with crop yield of 327 g/m^2^ and (k-15339) from Belarus with 295 g/m^2^; Tab2598 (k-44326) from Finland with 187 g/m^2^; and Karelskaya bezostaya (k-40579) from Karelia with 160 g/m^2^. Crop yields of reference cultivars were as follows: Moskovskaya 39,250 g/m^2^, Pamyati Fedina 410 g/m^2^, Zarya 380 g/m^2^, Moskovskaya 56,505 g/m^2^, and Nemchinovskaya 24,470 g/m^2^. Moskovskaya 39 and Mocкoвcкaя 56 cultivars had 1000-grain weights of 39.7–45.2 g; the values for Nemchnovskaya 24 were 39.7–45.2 g.

The data on high winter resistance of specific genotypes presented above show how far the breeding has advanced in this field. Plant breeders have developed not only winter-resistant cultivars that are well-adapted to the soil and climatic conditions of the Central non-Chernozem Belt, but also ones that combine high crop yield, disease resistance, and high grain quality [24]. 

Overcoming the negative correlation between winter resistance, crop yields, and grain quality has been a milestone achievement in the recent 30 years.

## 5. Regeneration Capacity in Spring after Poor Wintering

Mironovskaya 808 and the cultivars produced by the Research Institute of Agriculture of the Central non-Chernozem Belt such as Zarya (k-49916), Zarya 2 (k-54610), (Yantarnaya 50), Inna (k-62733), and Pamyati Fedina (k-62440) regrowing their root systems in spring showed high regeneration capacity throughout the whole research period (Figure 2). Among the foreign samples, this capacity was demonstrated by TAW3668.71 (k-55929) from Germany and Rmo (k-55220) from Poland.

## 6. Resistance to Soil and Atmospheric Drought

Throughout the whole research period, soil drought was only observed once in 1988. Grain filling was delayed by three weeks due to undersaturated soils.

Drought is a topical problem for a significant part of our country. In addition, renowned climatologists, both foreign and domestic [25,26], showed that the probability of this disaster will only increase in the decades to come. N.I. Vavilov reasonably believed [27] selection of drought- and heat-resistant crops and breeding of drought-resistant cultivars for various eco-geographical zones based on wide use of VIR collections were key instruments to combat droughts.

The first classification of drought-resistant agricultural plants cultivated in the USSR was developed by N.I. Vavilov and presented at the All-Soviet Union Conference on Drought Control held by the Academy of Sciences of the USSR and V.I. Lenin Academy of Agricultural Sciences in 1931 in Moscow [27].

N.I. Vavilov divided the multitude of plant species and genera into three groups from agricultural and environmental perspectives. Wheat belongs to the second group of plants with intermediate resistance. These were relatively drought-resistant plants of high variability capable of considerable yields in undersaturated conditions. Plants from this group are the most valuable from the cultivation perspective, as they occupy a major part (over 60%) of the cultivated areas. 

The study of the world assortment of agricultural plants showed that the most valuable starting material in terms of drought resistance was collected in our country. At present, the top-priority goals are as follows: mobilization of new forms of drought-resistant wheat plants from arid zones and from abroad; expansion of research on isolation of the genetic sources and donors of drought resistance genes, i.e., quicker introduction of new drought-resistant cultivars and hybrids of critical agricultural crops into production.

Throughout the whole period of research of the VIR winter wheat gene pool, strong atmospheric drought was observed in 1972 and 2010.

In 1972, it was only due to double intertillage that a small amount of soil moisture could be preserved. Wheat leaves almost completely dried by 26 June, and further dry matter accumulation was only possible through the root systems and stems that still remained green. As a result, grain filling still occurred, but their final sizes were 30–40% smaller than those under normal conditions.

Vegetation conditions in 2010 were rather unfavorable. The average temperatures exceeded the long-term normal value (16.4 °C) by 6.5°C (22.9 °C). The last rain was on 18 June and the subsequent rains were recorded as late as 3 September, the hydrothermal index being 0.8. Winter wheat collection samples achieved full ripeness on 10–15 July, i.e., almost a month before the optimal window. The drought resistance of collection samples was evaluated under conditions of severe atmospheric drought. A total of 500 samples was analyzed, and the following 10 were recommended for drought resistance breeding programs: Sv 71536 (k-54131) from Sweden; Taroz (k-64061), Taras (k-64065), Tarmer (k-64062), Tazit (k-64060), TAW 7032.74 (k-57008), and Severin (k-57222) from Germany; Gama (k-57581) from Poland; Ferrugineum 737.76 (k-54633) from the Moscow Region; and L-1749 (k-55971) from the Kursk Region. The outperforming samples showed crop yields of 53.5–68.0 cwt/ha compared to 48 cwt/ha of reference cultivar Moskovskaya 39 (1999) (average for 14 plots of 2 m^2^). In 1972, the 1000-grain weight of the reference cultivar Mironovskaya 808 was 36 g, 49 (out of 300) samples came close to reference values, and 12 exceeded those (Table 1). We evaluated the winter wheat samples for drought resistance, and the reference crop yield was 240 g/m^2^ in 1972, 372 in 1969 and 358 in 1970.

The vast majority of German samples showed lower crop yields than recognized varieties. The following samples showed crop yields matching those of reference cultivars: one sample of Steiners Strusi (k-44858) with a crop yield of 246 g/m^2^, four samples of Golland (k-39583) with 228 g/m2, Frankensteiner Brauner (k-40914) with 220 g/m^2^, Goldene Aue (k-40477) with 216 g/m^2^, and Stiegler 22 (k-26353) with 216 g/m^2^ came close to Mironovskaya 808 with 240 g/m^2^. The remaining samples, Heines Feverson (k-185), Shiriffs (k-1672), Kujavischer Weisser Kolben (k-6290), St 3876 50 (k-43054), Lohmanns Beseler III (k-26403), Cimbals Grossherzog V.Sachz (k-26205), Bensings Trotzkopf (k-26228), Berkners Continental Dickkopf 95 (k-26310), and Liessau (k-26354) showed crop yields of 202 to 210 g/m^2^.

The following samples stood out in terms of drought resistance and combination of qualities in the hyperarid year of 1972: Liessau (k-26354), a winter-resistant high-yield cultivar ripening a day later than the reference cultivar Mironovskaya 808.Heine Stamm 3256 (k-40864), a relatively short, standing winter resistant, coarse-grained, high-yield cultivar ripening at the same time as the reference.Lossdorfer Prasident Hanisch (k-40894), a short cultivar resistant to powdery mildew ripening at the same time as the reference.Heines 1751 (k-41245), a short, standing, winter resistant, high-yield cultivar ripening at the same time as the reference.Steiners Strusi (k-44858), a relatively short cultivar resistant to powdery mildew ripening at the same time as the reference with high yields in arid years.Bielers Edelepp (k-39737), a winter-resistant, coarse-grained, high-yield cultivar resistant to powdery mildew.38.12 (k-40105), a short, standing cultivar resistant to brown rust.Neuzucht 14/4 (k-40109), a short, standing, winter resistant, high-yield cultivar extremely resistant to powdery mildew and brown rust.Schindlers N.Z. (k-40472), a coarse-grained and relatively high-yield cultivar.Russe 991 (k-40858), a short-stem cultivar resistant to brown rust ripening at the same time as the reference.Stauderers Markus (k-35660), a winter-resistant, coarse-grained cultivar ripening a day later than the reference.k-39751 (T. compactum), a short, coarse-grained, winter-resistant cultivar resistant to brown rust.Halle 1020 (k-34063), a coarse-grained cultivar resistant to powdery mildew. 

The starting material of German winter wheat is of applied interest.

The problem of winter resistance and consistently high crop yields in the Russian Non-Black Earth Belt was successfully solved by the Mironovskaya 808 cultivar with moderate winter and frost resistance than some outstanding cultivars [28]. Production of highly winter-resistant cultivars in the Don River Region was possible as a result of crossing with winter forms of moderately winter-resistant wheat cultivars. Moskovskaya 39 has been in production for over 20 years. It is characterized by moderate winter resistance, high group tolerance to many diseases, and crop yields of 450–700 g/m^2^ in scientific crop rotation. This cultivar remains unrivaled in the region in the way it combines valuable agricultural qualities [29].

## 7. Conclusions

Abiotic stress factors have negative repercussions on wheat production worldwide. The most difficult challenge in wheat breeding is the necessity to combine a large number of valuable traits in one cultivar. It takes patience, hard work, and joint efforts of plant breeders, physiologists, phytopathologists, entomologists, and agricultural technologists to come close to an ideal wheat cultivar. These strategical objectives proposed by N.I. Vavilov are successfully solved at VIR and its experimental stations.

## Figures and Tables

**Figure 1 plants-10-02337-f001:**
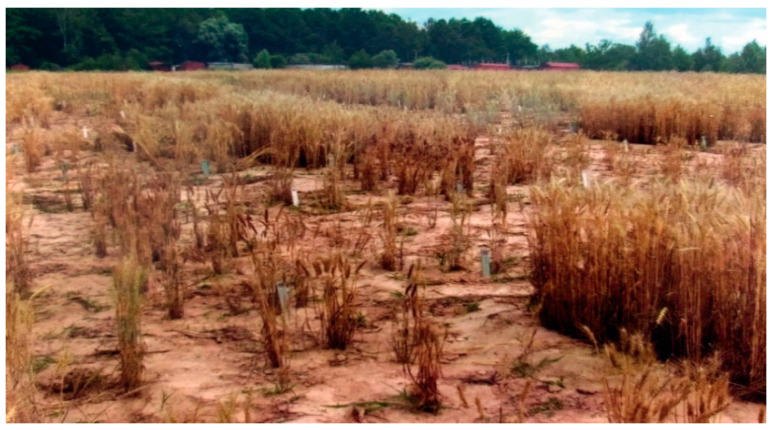
Surviving winter wheat cultivars after abiotic stress (winter hardiness).

**Figure 2 plants-10-02337-f002:**
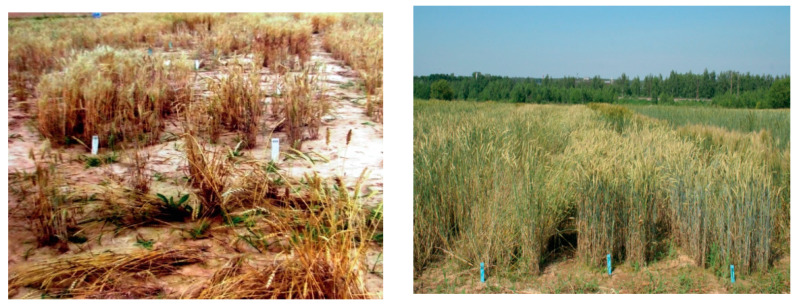
Wheat regeneration ability in spring after abiotic stress (winter hardiness).

**Table 1 plants-10-02337-t001:** German winter wheat samples with largest grains that stood out in terms of drought resistance (1972).

VIR Catalog No.	Name	1000-Grain Weight (g)
40469	*Heinrichs von Hindenburg*	40
40476	*Konkurrenzen von Meyer Wageninger*	38
26208	*HildebrandtsWeissweizen*	39
40467	*HeinrichsGelbkoernigerDickkopf*	39
45029	*Dippes Triumph*	38
43034	*Fanal*	37
40468	*HildebrandtsWeisserViktoria*	37
39737	*BielersEdelepp*	38
44973	*Skumstall*	37
40487	*Hallets Pedigree von Vilmorin*	37
44796	*Basta*	39
43920	*Mironovskaya 808 (reference)*	36

## Data Availability

For the first time, an analysis of a fifty-year study of the gene pool of winter wheat for resistance to abiotic stress is presented.

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
