# Peer review of "The Evaluation of Winter Wheat Adaptation to Climate Change in the Central Non-Black Region of Russia: Study of the Gene Pool Resistance of Wheat from the N.I. Vavilov Institute of Plant Industry (VIR) World Collection to Abiotic Stress Factors"

_plants, 2021, doi:10.3390/plants10112337_

Round 1
Reviewer 1 Report
This is a difficult paper to review; it is not a conventional style, but the subject matter - almost isolated observations when possible, not quite anecdotal, and over a long period of time. This also makes the paper not "cutting edge". These observations almost certainly cannot be repeated - certainly not by other researchers. However, I think it is important that these observations are recorded, and not buried in archives/oral histories. Therefore I suggest publication. I think Plants can help with the slight English editing required, although the paper is completely understandable as it stands.
Reviewer 2 Report
This work is of great importance for improving the resistances of wheat, as it present the results of long term evaluation of the world collection.
But the manuscript was not well organized, and need further improvement.
1. The abstract could not give clear message on the main results, such as the most varieties with drought resistance, etc.
2. Most of the paper were descriptions of the results, some tables or figures could be used to make the results more clear.
3. There were some descriptions on the other crops, unless really needed, I would suggest to remove them.
4. The author may provide some informations on the utilization of some distinguished materials, to support their values.
Reviewer 3 Report
I have read this manuscript with interest. However, it needs clarity in the methodology.
- Was this study carried out for 50 years consecutively? Who did it for 50 years? Or is this using data and modelling prediction?
- Please explain the rotation system clearly with the number of wheat genotypes planted clearly every year.
- What are the data collected and analysed?
- Which statistical tool was used for data analysis? Did you find any significantly different performance as compared to the checks?
Reviewer 4 Report
Τhe references are very old. It is necessary to complete with new articles
Round 2
Reviewer 2 Report
This manuscript summarized the evaluation results of the world winter wheat collection from N.I. Vavilov Institute of Plant Industry to abiotic stresses, especially on winter hardiness, which is important for the utilization of these germplasm. I have some comments below:
- in lines 145-146, the sentence of "the red clover ~ were lost" could be deleted.
- the released year of the most resistant cultivars might be added, such as those in lines 243-248, so that will be clear for the breeding progress.
- in lines 279-293, there were lots of informations unrelated with winter wheat, it should be removed, and focus on winter wheat, if there is no more data, the section of 6 Resistance to excessive soil saturation should be removed.
- in lines 295-326, those information were also not related with winter wheat, it should be removed.
- in lines 350-375, these could be as a summary of the most outstanding materials with good traits, and moved after line 387.
- please check thoroughly, to focus on winter wheat.
Reviewer 3 Report
Please include those points you indicated in your response letter in the methodology section of the manuscript. You didnt include them yet. It is critically important to include it.
